# Method for Applying Crowdsourced Street-Level Imagery Data to Evaluate Street-Level Greenness

**Xinrui Zheng [1,\*] and Mamoru Amemiya [2]**

[1] Doctoral Program in Policy and Planning Sciences, University of Tsukuba, 1-1-1 Tennodai,
Tsukuba 305-8573, Ibaraki, Japan

[2] Institute of Systems and Information Engineering, University of Tsukuba, 1-1-1 Tennodai,
Tsukuba 305-8573, Ibaraki, Japan

\* Correspondence: s2236007@s.tsukuba.ac.jp; Tel.: +81-29-853-5393

**Abstract:** Street greenness visibility (SGV) is associated with various health benefits and positively influences perceptions of landscape. Lowering the barriers to SGV assessments and measuring the values accurately is crucial for applying this critical landscape information. However, the verified available street view imagery (SVI) data for SGV assessments are limited to the traditional top-down data, which are generally used with download and usage restrictions. In this study, we explored volunteered street view imagery (VSVI) as a potential data source for SGV assessments. To improve the image quality of the crowdsourced dataset, which may affect the accuracy of the survey results, we developed an image filtering method with XGBoost using images from the Mapillary platform and conducted an accuracy evaluation by comparing the results with official data in Shinjuku, Japan. We found that the original VSVI is well suited for SGV assessments after data processing, and the filtered data have higher accuracy. The discussion on VSVI data applications can help expand useful data for urban audit surveys, and this full-free open data may promote the democratization of urban audit surveys using big data.

**Keywords:** street-level greenness; crowdsourcing; Mapillary; image filtering; XGBoost

## 1. Introduction

### 1.1. Background

The visibility of greenery or the amount of vegetation that can be viewed has been positively associated with various health benefits, such as restorative effects [1,2], patients' recovery from surgery [3,4], and physical activity promotion [5,6]. As a critical landscape element in urban environments, the visibility of street greenness is also believed to positively influence people's perception of the landscape, thus contributing to the attractiveness and walkability of streets [7–10]. Accurately measuring street greenness visibility (SGV) is crucial in providing statistical evidence to understand the impact of sensory functions of urban greenery on areas such as health and landscape perception. Furthermore, the efficiency and lower barriers for the measurements are also important for the applications in various scenarios, such as research on understanding the effects and greening programming assessments.

However, previous studies on the associations between urban greenery and the possible impacts on citizens use less SGV assessment; this is partly due to the difficulty of quantifying visibility, especially in data collection and value computation at a large scale. For instance, subjective measures such as questionnaire and audit surveys are often time-consuming and expensive [11–14]; the application of viewshed analyses is limited by the unavailability of high-resolution spatial data and computationally intensive measure [15,16]. These previous studies often focused on measures relying on more-easily accessed area-level data to grasp the greenery situation, using indices such as the green space ratio and normalized difference vegetation index [17–19]. However, these aerial-view

measurements do not fully capture all the greenery elements that humans experience (such as the green wall and shrubs under trees). Some statistical results further confirm the difference between results from the different perspectives [5,20,21], thus highlighting the particularity and indispensability of SGV assessments in urban greenery studies.

With the proliferation of street view imagery (SVI) platforms, advances in computer vision and machine learning, and availability of computing resources, SV-based greenness visibility measures have become increasingly common [22]; indeed, these machine-based methods have been proven to be in close agreement with human perception [23]. However, the current methods for measurements use limited types of SV data (Table A1), which has also restricted the applications of these methods. Most of the studies conducting greenness visibility measurements rely on data collected by certain companies or government agencies (which can be called top-down type data) [22,24]. Even though the top-down data collection method has advantages in controlling data quality and coverage, there are several barriers to data download and usage for users who are consumers rather than producers. For example, bulk downloading is often prohibited (e.g., Bing Streetside or Apple Look Around), and other SVI services like Google Street View (GSV) allow bulk downloads but usually charge a fee after a certain amount of free downloads.

Furthermore, obtaining historical imagery is not allowed by these SVI services, and they often have usage restrictions (some restrict the extraction of greenness information from imagery; e.g., [25,26]). These restrictions present issues when conducting research at a large scale or through temporal analyses. Thus, discussions on the possibility of applying open SV data in measurements are significant for the applications.

With the emergence of the Web 2.0 era, which has fostered the potential for individuals to contribute and access information through multiple resources [27], crowdsourcing has been applied to data collection in various domains, including volunteered geographic information [28], which involves large numbers of individuals in geographic information creation (e.g., OpenStreetMap or Wikimapia). The proliferation of location-aware devices, especially photography equipment, has also facilitated the collection of SVI in a short time and at a low cost and provides the opportunity to obtain fully free open data. For example, Mapillary (https://www.mapillary.com/ (accessed on 27 December 2022)) and KartaView (named OpenStreetCam until November 2020) (https://kartaview.org/landing (accessed on 27 December 2022)) collect panoramic or regular street-level photos contributed by their users with any GPS-enabled camera or smartphone devices from all over the world. Images from them are provided under CC BY-SA 4.0 license, meaning the data is free to use, even for commercial purposes. In this study, we assumed that these crowdsourced mapping data might be a potential in situ data source with a lower usage barrier for SGV assessments and other urban environmental audits.

For SV-based SGV measurements, the visibility value of a location is often defined based on the proportions of greenery pixels in images taken at those locations [29], thus creating high sensitivity to the environmental conditions while photographing (e.g., it is difficult to have a clear view of greenery at night or from an obstructed view). Therefore, researchers often choose to apply imagery data taken in well-controlled shooting environments (such as GSV, which usually collects pictures in good weather and during the day) to SGV assessment. Furthermore, to ensure these values correspond to the human perspective closely, researchers often try to cover the full view of the surrounding vegetation [28,30]. Alternatively, some surveys measure greenness from the usual viewing angles that people experience, as the behaviors of local residents can be regarded as a stereotype that does not change too much [31]. It is believed that their views generally follow the street layouts, and they do not look up into the sky frequently [32]. For instance, Ye et al. [32] used GSV pictures in four directions (the front, rear, left-hand, and right-hand) at the horizontal level to measure street greenery. Other cases are street greenery surveys by some of Japan's local governments, where front pictures (parallel to street segments) at the human-eye height are used for greenness calculations [33,34].

However, the variety of crowd data sources for volunteered street view imagery (VSVI) data collection has also led to the uncertainty of the photography method, which hinders the application of this novel type of data to SGV measurements or other urban audit surveys regarding proportional calculations. For instance, contributors who share photos may not struggle to avoid shooting environments such as low-light and obstructed views that are unsuitable for streetscape observations (Figure 1). Moreover, the camera angle is usually inconsistent, especially while walking and cycling, owing to the lack of professional photography equipment, leading to some pictures that are largely different from the general pedestrian view (e.g., pictures with too much ground or sky) (Figure 2). SGV values calculated based on the original VSVI data may not be able to reflect what people experience or guarantee comparability among locations.

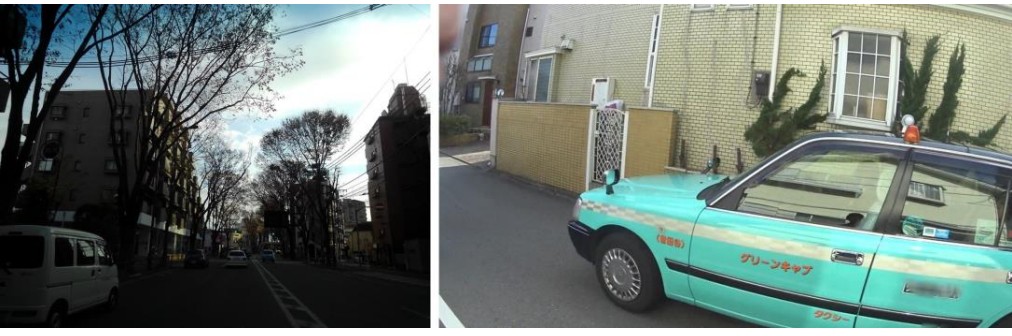

**Figure 1.** Examples of VSVI taken in a low-light environment (**left**) and with an obstructed view (**right**). Data are from the Mapillary platform.

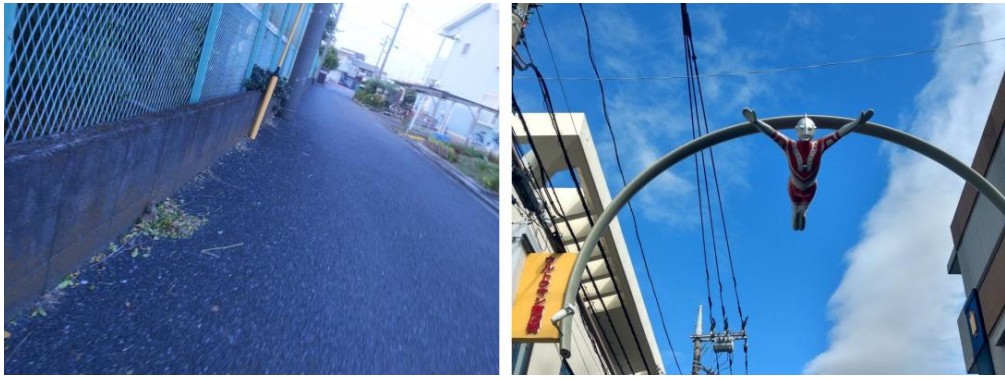

**Figure 2.** Examples of VSVI with too much ground (**left**) and sky (**right**). Data are from the Mapillary platform.

In this study, we assumed that SGV calculation only based on the appropriate photos from the original data would obtain more accurate results. However, image processing methods for this purpose have not been discussed in studies on SGV measurements and related urban audit surveys due to the predominant data type of panorama and images taken under certain requirements through field surveys. In addition, developing an automatic image filtering method would make the application more efficient. Although image classification methods have thrived in computer science through machine learning techniques, no application has been conducted for the specific purpose of images for streetscape monitoring. To improve the accuracy and efficiency of SGV measurements based on VSVI data, the application of automatic image classification techniques should be discussed.

*1.2. Objectives and Research Structure*

As the first attempt to apply VSVI in SGV measurement, this study examines the possible low accuracy problems from image quality issues by applying image classification

techniques to filter appropriate images from the original datasets. The results are expected to promote the development of efficient, high-accuracy, and low-barrier SGV assessment methods. Notably, a streetscape monitoring method that considers the ability to reflect the human perspective and the limitations of non-panoramic images with a single view (most VSVI are not panoramas) and includes pictures taken in an ideal environment, with the front view parallel with the street segment, and at the horizontal level (Figure 3). To do so, the following objectives were defined: (1) develop a framework to screen out qualified images from the VSVI dataset; and (2) evaluate the accuracy performance of the VSVI data in SGV measurements.

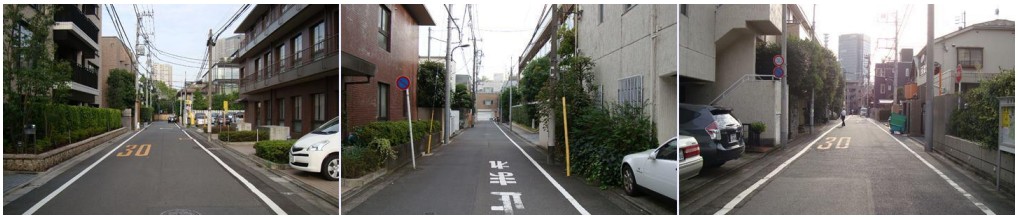

**Figure 3.** Examples of suitable images in SGV assessments. Reprinted with permission from Ref. [33]. 2017, Shinjuku City.

The remainder of this study is organized as follows. A description of the data sources and the applied research methods, including the image filtering method, are introduced in Section 2. Section 3 demonstrates the results of SGV measurements and the statistical analyses, which is followed by a discussion in Section 4 and conclusions in Section 5.

## 2. Materials and Methods

### 2.1. Overview of the Analysis and Study Area

This study proposed an image filtering method to improve the quality of VSVI data for SGV assessment and conducted an accuracy evaluation using a case study assessment. The imagery data used in this study were downloaded from the Mapillary platform, which is the first platform to provide crowdsourced SV services. Since the inception of Mapillary in 2014, user-submitted images have reached over 41 million in Japan as of 2020 [35]. Shinjuku was chosen as the study area considering the availability of Mapillary imagery data (which is incomplete in Japan) and the existing reference SGV data for accuracy evaluations. Shinjuku, located in Tokyo, is regarded as one of the hotspots of Mapillary activity. In terms of reference data, while GSV is widely believed to provide reliable imagery data in SGV assessment, using applications "to analyze and extract image information" has recently been prohibited recently [25]. This limits the application, particularly in districts where there are no fair use exceptions for copyright infringement (including the districts in Japan). The local government in Shinjuku conducted greenness visibility surveys twice (1984 and 2016) to understand the current situation and the changes in greenery along streets to create the basic data for green policy planning. The images used in these surveys were captured under strict and consistent criteria that are similar to the definition of suitable images in the current study [33]. The detailed methods, results, and imagery data of the latest survey have been published and can be used as reference data to accurately evaluate the results from the VSVI data.

Shinjuku is a centrally located ward in Tokyo (Figure 4) with an area of 18.22 km$^2$ and a population of 340,877 (as of 1 April 2022). This city is dominated by residential and commercial land use, and the area around Shinjuku Station is Tokyo's largest commercial and amusement area with many high-rise buildings on the west side (Nishi-Shinjuku). Shinjuku has a green coverage of 17.98%, which is primarily provided by parks, schools, and public facility areas. The major green spaces include the Shinjuku Gyoen Garden and Meiji Jingu Shrine Gaien (Outer Garden) in the South, Shinjuku central park in Nishi-Shinjuku, and the Toyama Park in the North (Figure 4). According to the SGV survey results, the mean value is 18.12%, which ranges from 14.38% to 22.69% across districts [33].

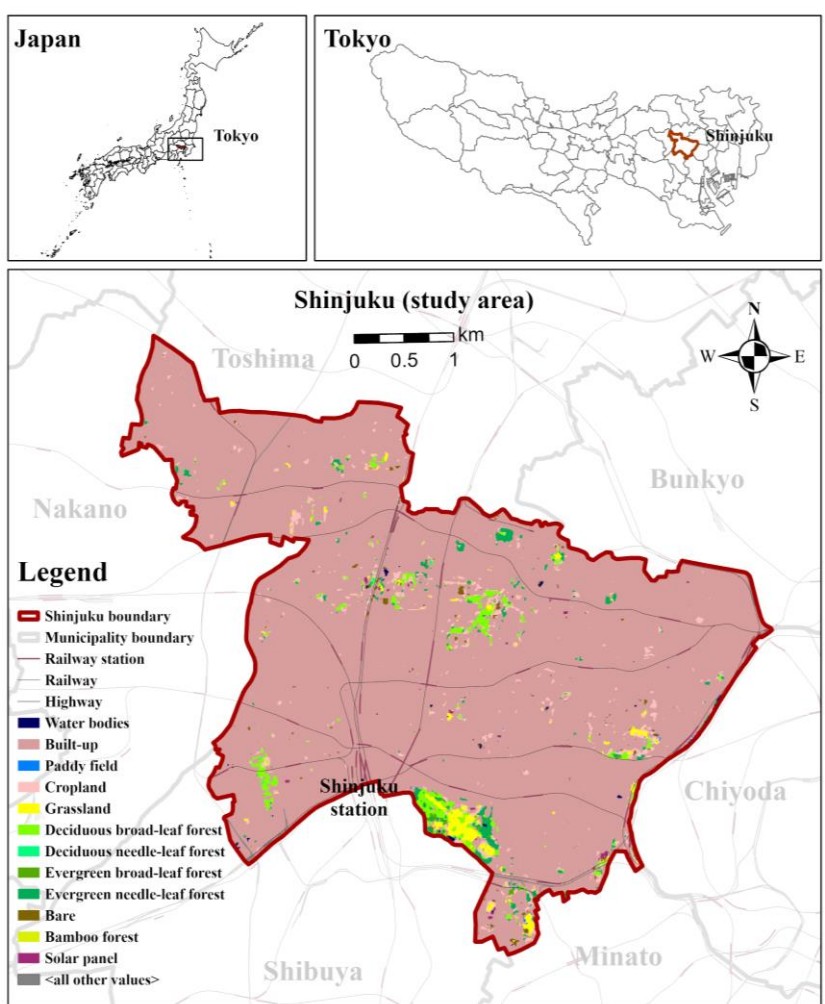

**Figure 4.** Location and land cover map of the study area. Data were provided by the Japan Aerospace Exploration Agency's high-resolution land-use and land-cover map of Japan [36].

### 2.2. Data Description and Extraction

2.2.1. Mapillary Imagery Data

The metadata of Mapillary imagery can be accessed via Mapillary's official application programming interface (API), through which attributes such as the *image id*, *coordinates*, *is_pano*, *captured_at*, and *url* for the downloaded image can be retrieved. By setting the parameters of the API query, Mapillary allows developers to search for images close to specified locations by inputting the coordinates of that locations and search distance. The coordinates of sample intersections applied in this study were determined with the road center line data from a digital map published by the Geospatial Information Authority of Japan generated in 2019. Data used in this study were images of less than 10 m (considering the road width and GPS error) from the sample intersections in Shinjuku until 3 November 2021. The normal non-panoramic images are predominant in the Mapillary dataset due to the lack of widespread availability of panoramic cameras for the crowd. For the data preprocessing, we first screened out panoramic images, making use of the attribute *is_pano* (if it is a panoramic image) as this study focuses on normal images. Following this, we also excluded images captured within fall foliage seasons, which may affect the discussion on greenery assessment accuracy, using the attribute *captured_at*. After the preprocessing, the final dataset contained 1049 photos from the original Mapillary data set.

2.2.2. Reference Data for Accuracy Evaluation

A reference SGV value dataset with high accuracy is needed to evaluate the accuracy of survey results based on the aforementioned crowdsourced data. In Shinjuku, street imagery data for SGV calculation were collected through field surveys from 27 September to 7 October 2016. During these field surveys, the street pictures were always taken in the direction of the roads, with a 35 mm lens (35 mm equivalent focal length) and from a height of 1.5 m. For the sample locations, the Shinjuku area was divided into equal size grids (230 m × 230 m), and the intersections nearest the nodes of these grids were selected as sample points. The results published in the report are based on 937 photos taken from 287 intersections in every direction of roads from these intersections. The percentage of greenery pixels within each picture was calculated with the software Photoshop and defined as the SGV value, and the mean SGV value for all images from the same intersections was used as the SGV value for that intersection. The report published the SGV values for the sample intersections, imagery data, and calculated values for each image [33].

*2.3. Image Filtering Method*

Images uploaded to the Mapillary platform can be taken by any user, using any method, at any time, and from any location (even indoors), resulting in a significant large difference in image quality. Owing to the complexity and variety of urban environment, features related to the criteria for suitable images defined in this study (Section 1.2; such as the perspective, light condition, shooting object, and the presence of obstacles) are difficult to fully extract automatically without large amounts of training data. Thus, we chose to achieve an automatic image screening process by specifying some related image features of images using multiple computer vision techniques and classifying them using ensemble methods. More specifically, the Extreme Gradient Boosting (XGBoost) method, an ensemble learning method developed by Chen and Guestrin [37], was used in this study due to its effective performance in many machine learning tasks. Figure 5 depicts an overview of the image filtering methods, which are divided into three sections: setting, image labeling, and model training.

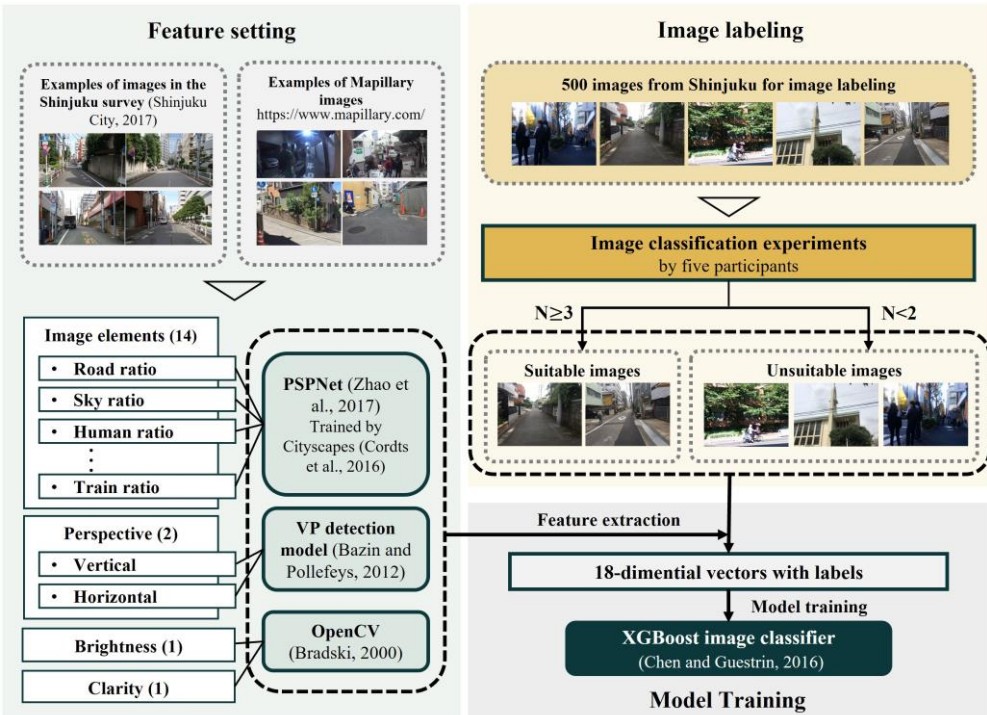

**Figure 5.** Overview of the method [33,37–41].

Feature setting is based on the observation of Mapillary pictures and shooting methods adopted by the Shinjuku government [33]. First, to evaluate the greenness visibility of the streetscape, the places photographed are generally along ordinary roads other than highways, railways, and tunnels, or in some indoor places. As the distinct difference between these pictures and the desired ones in the composition of image elements (such as the rarely seen sky elements in pictures taken indoors and inside tunnels and train elements in pictures of railways), the semantic segmentation method (we used PSPNet model [38]) trained by Cityscapes dataset [39] in this study) to identify the pixels' object classes was used for the image elements extraction. For the photographing method, the Shinjuku government chose to take pictures aimed in the direction of the roads and level with the horizon, which can be distinguished by whether the position of the vanishing point (detected by a 3-line RANSAC VP model [40]) is close to the center point in a picture. Notably, the shooting height (1.5 m) and the angle of view were ignored due to the nuanced differences between the different values. Moreover, imagery data from the Shinjuku government demonstrate the requirements for sufficient-light situations, which can be reflected in the brightness value (by OpenCV [41]) and an unobstructed view that can be seen from the obstacle elements. Finally, we added the clarity feature (by OpenCV) for the commonly seen blurry pictures taken by users, especially while walking and cycling.

Before the image classifier training, it is necessary to establish an imagery dataset with specific classification results. Labeling unsuitable pictures for those taken along some unwanted types of roads (e.g., highways and tunnels) may be easy based on the visual information but not for those depending on some qualitative characteristics without clear thresholds; for instance, it is hard to determine what level of darkness would make an image unsuitable. To resolve this, the image classification was based on the existing perceivable differences in quantifying streetscape elements between the pictures to be labeled and the hypothetical suitable pictures capturing the same streetscape. To achieve objective results, experiments were conducted involving multiple people to select results from most of the participants. Five participants from the University of Tsukuba were selected using the snowball method for efficiency; notably, none were from the research group. These experiments took place from 2–3 November 2021, for each participant. To ensure the understanding of participants, the experiments were conducted offline using an introduction file (Appendix B) containing some examples of suitable images, which are from the Shinjuku survey, and some unsuitable representative images from the Mapillary platform (no duplication with images from the training data) to explain the aims of classification. Data for the experiments were 500 pictures randomly selected from an imagery dataset downloaded via the Mapillary API and the locations of 200 random intersections in Shinjuku (72 of them with available Mapillary data; Figure 6). Participants were asked to divide these images into two folders on a computer. Finally, the pictures were grouped as suitable if three or more people labeled them so. Among these 500 pictures, 20 were removed due to high similarity, and then the labeled dataset consisting of 480 pictures was used as a training set for XGBoost classifier training.

This study built the XGBoost model using Python software along with the "*sklearn*" package and "XGBoost" package. The training set was randomly split into training (80%) and testing (20%) data sets. The parameters of XGBoost were tuned using the Grid Search for model optimization, and the final parameters were determined while reaching a sufficiently high precision. The model we built contained 100 decision trees (*n_estimator*), the learning rate (*learning_rate*) was 0.03, and the minimum loss reduction required to make a further partition (*gamma*) was set to 0. In addition, the minimum sum of instance weight needed on a leaf node for a further partition (*subsample*) was set to 0.8. The above parameters helped reduce model complexity and prevented overfitting.

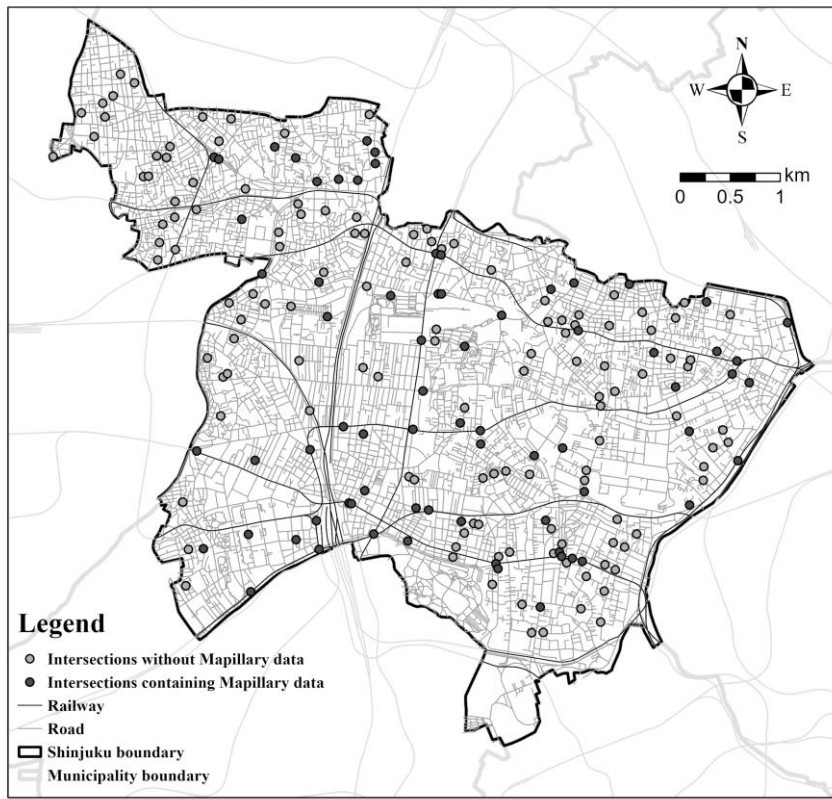

**Figure 6.** Distribution of 200 randomly selected intersections for training data retrieval.

*2.4. Accuracy Evaluation*

To examine the accuracy for SGV measurements using Mapillary data, this section compares the performance of SGV survey results with the reference data (Section 2.2.2) using the original and filtered Mapillary datasets, respectively. The comparison was conducted at the level of road direction rather than the intersection, which is due to the incompleteness of Mapillary images. The study intersections were determined with the help of point distribution maps and the available imagery data in the report to ensure the investigation points were consistent with the reference data. Among the 287 intersections, nine were excluded as they could not be located, one was excluded due to a lack of complete SGV values for each image, and five were excluded to avoid duplication with the intersections for the training set. The images were assigned road directions where the difference between the angle of the image and the road is minimum, making use of the compass angle retrieved from the image metadata. The SGV value for each Mapillary image was determined with the PSPNet model trained by the Cityscapes dataset, and the average of SGV values of images viewed from the same road direction was defined as the road value. The following statistical evaluation metrics were used to assess the accuracy of the SGV results: Spearman's ranking correlation coefficient (*r*), root-mean-square error (RMSE), and relative bias (RB; to measure the tendency of overestimation or underestimation).

**3. Results**

*3.1. Image Classification Method*

In total, 18 image features were extracted with the PSPNet model, OpenCV software, and VP detection model reflecting the characteristics of photographing place, environment, and method; these features were selected for the classifier construction and are illustrated in Table 1. A 480-picture dataset consisting of 214 suitable and 266 unsuitable pictures was formalized as 18-dimensional feature vectors and used as training data for XGBoost classifier implementation. The model was optimized using Grid Search. An out-of-sample set of 20% dataset (96 pictures) was kept to test our model's predictive power, of which

42 were suitable and 54 were unsuitable. As the confusion matrix illustrated in Table 2 demonstrates, the model diagnosed 36 suitable and 60 unsuitable pictures, which achieved an accuracy of 87.5%, precision of 91.7%, and recall of 78.6%.

**Table 1.** Image features for classification.

| Features | Description | Extraction Method |
|---|---|---|
| 1. Image elements (the ratio of roads, sidewalks, buildings, walls, fences, sky, riders, cars, buses, trains, motorcycles, and bicycles within the images) | To distinguish the photographing place<br>To recognize obstacles | PSPNet model trained by Cityscapes |
| 2. VP position (vertical and horizontal distance to the center point of the image) | To ensure images are level with the horizontal position and aimed at the road direction | 3-line RANSAC VP detection model |
| 3. Brightness (brightness value) | To avoid pictures taken in low-light situations such as night and rainy days | OpenCV |
| 4. Clarity (clarity value) | To recognize blurry pictures | OpenCV |

**Table 2.** Confusion Matrix for XGBoost model.

| | | Predicted Class | |
|---|---|---|---|
| | | Suitable | Unsuitable |
| Actual class | Suitable | 33 | 9 |
| | Unsuitable | 3 | 51 |

### 3.2. Accuracy Evaluation

The coordinates of a total of 272 intersections in Shinjuku with reference SGV data were input into the Mapillary API query, and 2122 photographs were retrieved from 96 intersections that make up the original Mapillary imagery dataset. These images were then formalized as 18-dimensional feature vectors, which were then input to the established XGBoost classifier. After removing panoramas and pictures taken in the fall foliage seasons, a total of 1082 images remained for further analyses, and these images were matched with 123 road directions based on the compass angle attribute. However, 213 (19.7%) of these images were found with the wrong compass angle and matched to the wrong road direction, possibly due to GPS sensor issues on devices (all these images are from only two power users). The road direction these images assigned were revised, referring to imagery used in reference data. In addition, we also removed five roads (matched with 33 images) with reference images that were taken without following rules (e.g., taking pictures in front of greenery rather than the center for the reason of traffic safety). After these steps, 1049 images were left for further study consisting of the original imagery dataset, and 282 were identified as suitable images with the developed image classification model to make up the filtered Mapillary imagery dataset.

The mean SGV value of Shinjuku based on the original Mapillary dataset is 10.72% (SD = 10.91%, $n = 114$) and ranges between 0.00% and 52.89%. Meanwhile, the mean value based on the filtered Mapillary dataset is 12.81% (SD = 12.03%, $n = 75$) and ranges between 0.00% and 52.89%. The reference SGV values from the survey report demonstrate a mean value of 14.66% (SD = 14.68%, $n = 114$) and a range from 0.00% to 55.47%. The frequency distribution of these results is displayed in Figure 7. Generally, the results based on the three datasets do not have a normal distribution and demonstrate a higher frequency of values close to 0. After the image filtering process, the SGV values demonstrate a lower frequency of samples with low values (less than 10%), which is closer to the distribution of reference data.

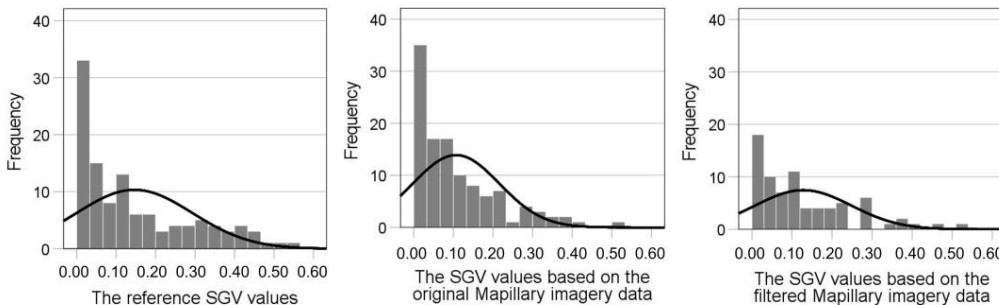

**Figure 7.** Frequency histograms of the SGV values.

Figure 8 shows two scatterplots of the two Mapillary datasets and the reference data, indicating that the deviation values decreased after the image filtering method and these SGV values were closer to the 45° line. Two logarithmic scatterplots (Figure 9) were used to better visualize the data distribution of values close to zero. SGV values calculated using the original Mapillary imagery dataset had an RMSE of 0.11, which slightly improved after image filter processing (0.09) at a rate of 22.4%. The RB result shows that the original Mapillary data underestimated the SGV values by 26.89%, and the underestimated trend decreased when using the filtered data (RB = −16.77%).

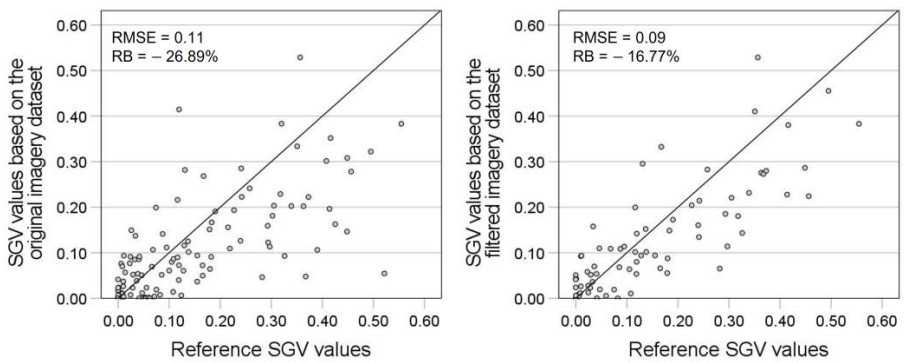

**Figure 8.** Scatterplots of SGV values from the Mapillary imagery dataset and the reference SGV values before (**left**) and after (**right**) image filtering.

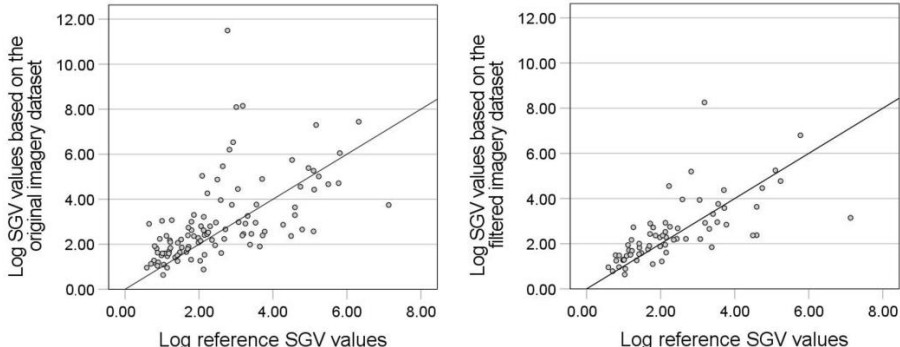

**Figure 9.** Log-log scatterplots of SGV values based on the Mapillary imagery dataset and the reference SGV values before (**left**) and after (**right**) image filtering.

Spearman's rank correlation coefficients were used to measure agreement between these three sets of SGV values, and the results are illustrated in Table 3. The highest correlation (0.919, $p < 0.01$) appears between values based on the original and filtered Mapillary data. A strong correlation (0.740, $p < 0.01$) was found between the original Mapillary values and the reference values, and the performance was further improved (0.829, $p < 0.01$) after the image filtering process.

**Table 3.** Spearman's rank correlation of SGV values.

|  |  | 1 | 2 | 3 |
|---|---|---|---|---|
| 1 | Values based on the original Mapillary dataset (*n* = 114) | 1 |  |  |
| 2 | Values based on the filtered Mapillary dataset (*n*= 75) | 0.919 ** | 1 |  |
| 3 | Reference values (*n* = 114) | 0.740 ** | 0.829 ** | 1 |

** = $p < 0.01$.

## 4. Discussion

In this study, we proposed using VSVI data to assess the visibility of greenery along streets. Considering the low-quality pictures in crowdsourced data, which may affect the ability to reflect human perspective accurately, an image filtering model applying multiple computer vision techniques and XGBoost was developed to process the original VSVI data. Finally, the accuracy of the SGV values based on the VSVI data was evaluated using the reference data published by the local government, and the application of the filtering process demonstrated a modest improvement in SGV monitoring accuracy.

Our automatic image filtering method is a feasible and efficient tool to distinguish suitable street-level images for SGV surveys. In this image classification task, we made requirements for some attributes (such as shooting environment and method) based on the streetscape perception ways of citizens. Considering the diversity of the urban environment, which made it difficult for models to discover the key features, related variables were specified manually with some computer vision techniques. An XGBoost model was applied for the high effectiveness in dealing with a limited amount of training data and the advantages of being easy to use and efficient. Consequently, the established image filtering model achieved a high precision of 91.7%, even with limited training data (500 pictures). The model can also be used in image preprocessing for other urban audit tasks regarding streetscape element proportion calculations, such as the sky ratio. Furthermore, model design ideas have reference significance for classifying images with complex requirements and too much distracting information.

The results of our research demonstrate that VSVI data can be useful in SGV assessments, and the results can have a relatively higher accuracy after appropriate data processing. Unlike the commonly used panoramas used in previous studies [5,28], most of the VSVI are normal pictures with a fixed view angle of streets that require unique data processing methods. Furthermore, the variety of the data sources from the crowd casts doubt on the reliability of the survey results. After seasonal filtering and road matching, the assessment results show a slightly smaller RMSE (from 0.11 to 0.09), a moderately lower RB (from −26.89% to −16.77%), and a slightly stronger correlation (from 0.740, $p < 0.01$, to 0.829, $p < 0.01$), confirming the high accuracy of the results. The difference between the VSVI and reference data results may be explained by unavoidable factors, such as plant growth in different months (Figure 10) and years (Figure 11), as well as different shooting positions (Figure 12).

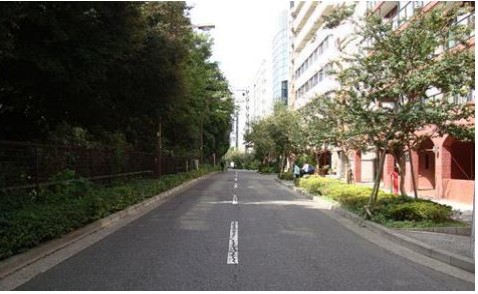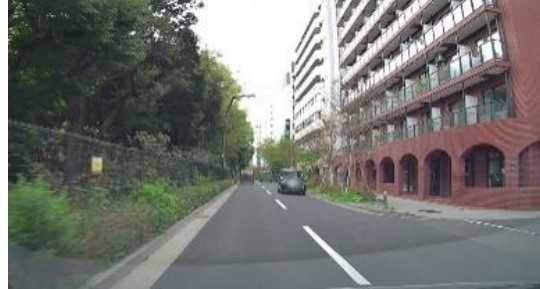

**Figure 10.** Comparison of a reference picture taken in early October (**left**) and a Mapillary picture taken in April (**right**). Data are from the Mapillary platform.

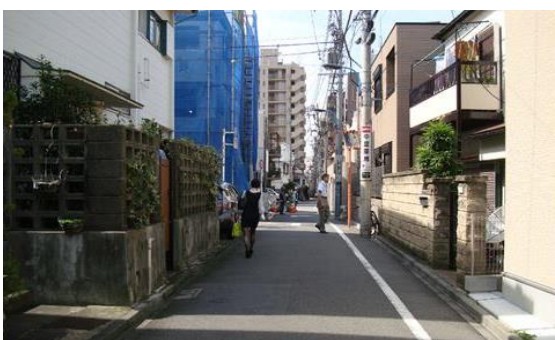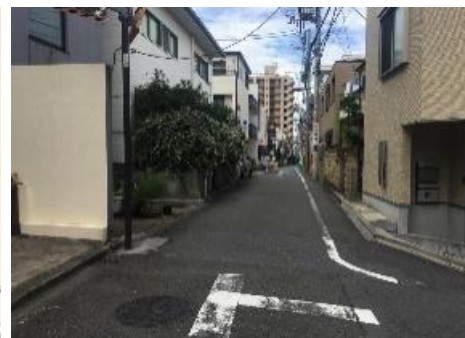

**Figure 11.** Comparison of a reference picture taken in 2016 (**left**) and a Mapillary picture taken in 2021 (**right**). Data are from the Mapillary platform.

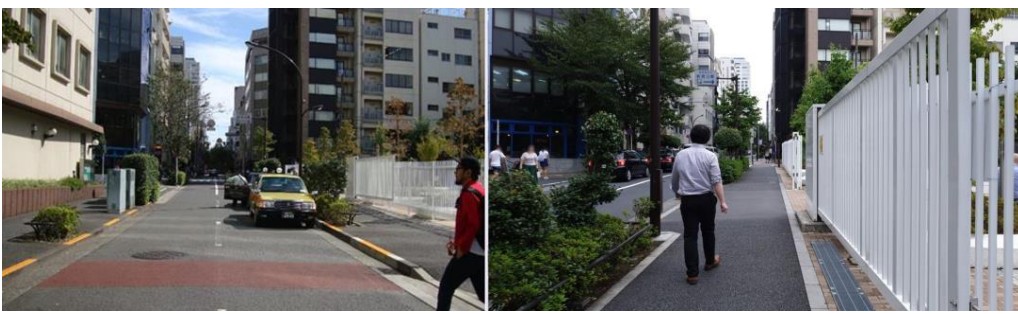

**Figure 12.** Comparison of a reference picture of a road (**left**) and a Mapillary picture of a pedestrian path along the same road (**right**). Data are from the Mapillary platform.

VSVI has received increasing attention from the research domain in recent years [35,41–45] and has been used in various surveying tasks such as road information extraction [46,47], building observation [48], and crop monitoring [49]. Using full-free open VSVI data in SGV surveys can help to expand the types of useful data in streetscape monitoring research and facilitate the democratization of urban audit surveys using big data. This largely facilitates urban audit studies particularly in districts where the use of traditional top-down data is restricted. Verifying the possibility of using alternative open data greatly expands the data users to some groups, such as the government for evidence-based policy planning, private companies for application development, and individuals for private use.

However, there are some issues that need to be resolved in the development of the image filtering method, as well as some limitations in accuracy evaluation. The first concern is about the image filtering method. The feature extraction process involves the use of multiple models, thus increasing the workload and resulting in lower accuracy due to accumulated errors. Furthermore, the areas used in this study for training data retrieval and assessment applications were both Shinjuku, which may lead to higher accuracy due to the urban environment's similarity. Future research should look into the generality of this image classification model. The accuracy evaluation has limitations due to the choice of reference data and the size of the data volume. The reference SGV data for this study are based on images taken under rules formulated by the Shinjuku government at a specific point in time, rather than the widely used gold standard. Future research will need to look into comparisons with more recognized data. Furthermore, due to the limited number of sample sites with reference data and the lack of Mapillary images in some of these sample sites, the analysis of this study was limited to 114 samples. This calls the accuracy assessment's credibility into question. Future research should assess the accuracy of using VSVI data with a larger sample size.

Finally, while this study confirms the reliability of VSVI data in streetscape monitoring studies, it also identifies some limitations that may hinder its dissemination. First, the incompleteness of the VSVI data affects the application. Only 96 intersections (49 after the

filtering process) out of 272 have available data, and only a small number of intersections have pictures from all road directions. Even though the amount of Mapillary imagery data in Japan has surpassed 40 million [35] and is growing at an exponential rate, complete coverage of roads and directions remains difficult for VSVI data contributed by the crowd, even in some data-rich countries [44]. The evaluation of the completeness of this dataset and discussion of the strategy for improving data coverage are critical to the promotion of its application. The uncertainty in the accuracy of image attributes was discovered as a second issue. During the road matching process, we discovered some errors in the orientation attributes of these crowdsourced images, which complicate data preprocessing (the orientation of images was checked manually in this study). Even though the errors only occurred in images submitted by two users, these images accounted for nearly 20% of the retrieved dataset because these two were power users. To improve the reliability and efficiency of its application, methods to check and revise the orientation and other related attributes of crowdsourced imagery data should be discussed in future research.

## 5. Conclusions

Facing the usage restriction of the commonly used SVI data, this study applied the representative VSVI data of Mapillary data to SGV assessment and attempted to improve accuracy by quality improvement via an image filtering model. The study area for data retrieval was Shinjuku, Japan, and with the help of computer vision science and machine learning techniques, a high-quality model was developed using limited training data. Even though some shortcomings of the VSVI data (incompleteness and attribute errors) were found, the results demonstrated that the VSVI data are qualified for assessing street greenery, and the filtering method can further improve the accuracy of the assessment results.

As a new product of the Web 2.0 era, the VSVI database has quickly become valuable worldwide and is expected to flourish in the future with the development of the digital city. Even though the differences with reference data seem to lower the correlations in this study, the VSVI data were also excellent at delivering multi-perspective and multi-temporal environmental information, owing to the variety of data sources from the crowd. For a long time, the widely used SVI data (e.g., GSV, Baidu Total View) in urban audit studies have been providing the setting view of the streets (mostly from the car views) and restricting users from fetching historical data. The novel type of SVI data could provide the opportunity for more complex urban audit tasks involving perceptive and temporal characteristics. This study can be viewed as an extension of available SVI data in the field of urban studies and provides building blocks for person-centered and dynamic streetscape monitoring in the future.

**Supplementary Materials:** The following supporting information can be downloaded at: https://www.mdpi.com/article/10.3390/ijgi12030108/s1, Figure S1: suitable images; Figure S2: unsuitable images; Table S1: prediction result; Table S2: feature data; Table S3: imageid lookup table; Model S1: image filtering XGBoost model; Code S1: OpenCV; Code S2: PSPNet; Code S3: vpdetection.

**Author Contributions:** Conceptualization, Xinrui Zheng and Mamoru Amemiya; methodology, Xinrui Zheng and Mamoru Amemiya; software, Xinrui Zheng and Mamoru Amemiya; validation, Xinrui Zheng and Mamoru Amemiya; formal analysis, Xinrui Zheng and Mamoru Amemiya; investigation, Xinrui Zheng; data curation, Xinrui Zheng; writing—original draft preparation, Xinrui Zheng; writing—review and editing, Xinrui Zheng and Mamoru Amemiya; visualization, Xinrui Zheng; supervision, Mamoru Amemiya; funding acquisition, Mamoru Amemiya All authors have read and agreed to the published version of the manuscript.

**Funding:** This work was supported by MEXT KAKENHI Grant Number 21H01558.

**Data Availability Statement:** The data presented in this study, which are available in the Supplementary Materials, consist of (1) a "Feature data" file; (2) an "imagefiltering_xgboostmodel" model file; and (3) the "prediction_result" data files, which include a "prediction_result" file and the classified Mapillary images (which are released under a CC BY-SA license); (3) "source python code" python files, which include an "OpenCV" file, a "PSPNet" file, and a "vpdetection" file; and (4) an "imageid

lookup table" file matching id prepared in this study and in the Mapillary platform. Further data that support the findings of this study are available from the corresponding author on request.

**Conflicts of Interest:** The Mapillary images used in this study are all licensed under the Creative Commons Attribution-ShareAlike 4.0 International License.

## Appendix A

**Table A1.** Summary of the Literature Review.

| Authors | Year | Study Area | Purpose | Imagery Data | Panorama | Data Collector |
|---|---|---|---|---|---|---|
| Yang et al. [21] | 2009 | Berkeley, United States | To develop the Green View Index (GVI) to evaluate the visibility of urban forests | - | No | researcher |
| Li et al. [29] | 2015 | New York, United States | To propose a modified GVI formula using GSV images | GSV [1] | Yes | company |
| Li et al. [50] | 2015 | Hartford, United States | To explore the distribution of street greenery and its association with residents' socioeconomic conditions | GSV | Yes | company |
| Long and Liu [51] | 2017 | 245 major Chinese cities | To propose an automatic method to determine street greenery and analyze the distribution of street greenery | TSV [2] | Yes | company |
| Jiang et al. [18] | 2017 | The Midwestern United States | To assess associations among two remotely sensed and three eye-level tree cover density measures | - | Yes | researcher |
| Seiferling et al. [52] | 2017 | New York and Boston, United States | To test a novel application of computer vision to quantify urban tree cover at the street-level | GSV | Yes | company |
| Dong, Zhang, and Zhao [53] | 2018 | Beijing, China | To quantify street greenery in study area, analyze the relations with road parameters, and compare the visual greenery of different road types | TSV | Yes | company |
| Lu, Sarkar, and Xiao [54] | 2018 | Hong Kong, China | To develop methods and tools to assess the availability of eye-level street greenery and investigate the effect of street-level greenery on walking behavior | GSV | Yes | company |
| Villeneuve et al. [6] | 2018 | Ottawa, Canada | To assess associations between greenness, walkability, recreational physical activity, and health (comparing the NDVI with the GSV measure of vegetation) | GSV | Yes | company |
| Zhang and Dong [55] | 2018 | Beijing, China | To investigate the impacts of street visible greenery on housing prices | TSV | Yes | company |
| larkin and Hystad [19] | 2019 | Portland, United States | To evaluate GSV-based green space exposure measures as new approach for health research | GSV | Yes | company |
| Lu [56] | 2019 | Hong Kong, China | To assess both the quantity and quality of street greenery and investigate the association between them and physical activity | GSV | Yes | company |
| Yang et al. [57] | 2019 | Hong Kong, China | To examine the associations of urban greenery and older adults' physical activity | GSV | Yes | company |
| Ye et al. [58] | 2019 | Shanghai, China | To measure the potential economic effect of street greenery | BTV [3] | Yes | company |
| Ye et al. [32] | 2019 | Singapore | To propose an approach for quantifying the daily exposure of urban residents to eye-level street greenery | GSV | Yes | company |
| Chen et al. [59] | 2020 | Shenzhen, China | To explore the influence of greening factors on the use of shared bicycles | TSV | Yes | company |

**Table A1.** *Cont.*

| Authors | Year | Study Area | Purpose | Imagery Data | Panorama | Data Collector |
|---|---|---|---|---|---|---|
| Kumakoshi et al. [61] | 2020 | Yokohama, Japan | To propose an improved greenery visibility indicator (standardized GVI) and quantify the relation between sGVI and other green metrics | GSV | Yes | company |
| Tong et al. [62] | 2020 | Nanjing, China | To assess street greenery using multiple indicators | TSV | Yes | company |
| Wang et al. [63] | 2020 | Shenzhen, China | To explore the relationship between eye-level greenness and cycling behaviors | TSV | Yes | company |
| Wu et al. [64] | 2020 | Beijing, China | To investigate the effect of street greenery on active travel considering road classification | BTV | Yes | company |
| Zang et al. [65] | 2020 | Hong Kong, China | To explore the relationship between urban greenery and walking behaviors of older adults | BTV | Yes | company |
| Ki and Lee [66] | 2021 | Seoul, Korea | To examine GVI (the difference with traditional greenery variables) and explore its associations with walking activities | GSV | Yes | company |
| Li [67] | 2020 | New York, United States | To map and analyze the spatial distribution and temporal change in the GVI | GSV | Yes | company |
| Xia et al. [68] | 2021 | Osaka, Japan | To develop a method to determine the greenery amount of street view images and propose the Panoramic View Green View Index for measuring the visible street-level greenery | GSV | Yes | company |
| Yang et al. [69] | 2021 | Hong Kong, China | To examine the effects of streetscape greenery on the walking behavior of older adults | GSV | Yes | company |
| Yang et al. [70] | 2021 | Hong Kong, China | To develop a novel method to assess both the quantity and quality of park greenery from eye-level photographs of parks and explore the associations with park usage | - | No | researcher |
| Zhang, Tan, and Richards [71] | 2021 | Singapore | To examine the associations of different indicators of urban green spaces with health | GSV | Yes | company |
| He et al. [72] | 2022 | Shanghai, China | To examine the complex relationship between urban density, urban greenery, and older people's life satisfaction | BTV | Yes | company |
| Xue et al. [73] | 2022 | Guangzhou, China | To introduce Visible Difference Vegetation Index for GVI calculation and explore the spatial distribution of street greenery in Guangzhou | BTV | Yes | company |

[1] Google Street View. [2] Tencent Street View. [3] Baidu Total View.

## Appendix B

*Image Classification Experiment*

Not all pictures are suitable for streetscape monitoring in the crowdsourced street-level imagery dataset. Before implementing an effective image classification model for quality improvement, it is necessary to create training data containing images that are labeled objectively. This experiment was designed to label images objectively and classify images from multiple participants. Please help us to divide the provided 500 images into two folders, suitable and unsuitable. The requirements for suitable images for streetscape monitoring are shown below:

1.  Pictures of ordinary roads;
2.  Pictures with the front view parallel with the street segment and taken at the horizontal level;
3.  Pictures taken in an ideal environment for streetscape elements observation.

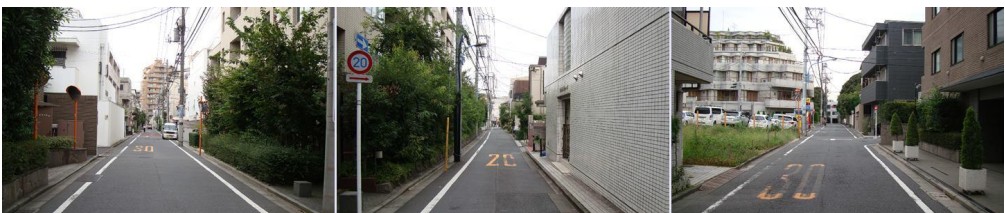

**Figure A1.** Examples of suitable images [33].

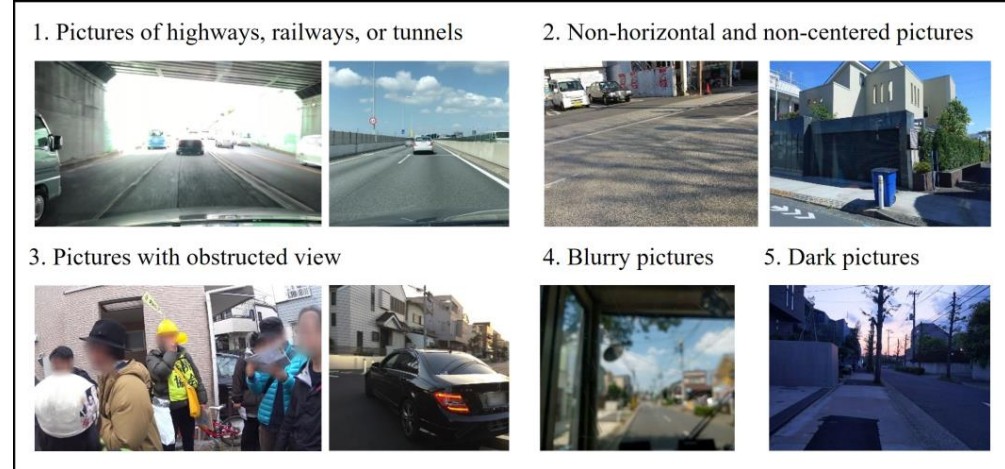

**Figure A2.** Examples of unsuitable images from the Mapillary platform. Data are from the Mapillary platform.

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
