# Peer review of "Method for Applying Crowdsourced Street-Level Imagery Data to Evaluate Street-Level Greenness"

_ijgi, doi:10.3390/ijgi12030108_

Round 1

Reviewer 1 Report

Thanks for inviting me to review this paper on using freely available mapillary images to model greenness visibility on streets. This interesting study develops valid arguments about how free data can supplement existing proprietary data with some filtering. I think the paper has merit and can be published. However, I recommend the author should argue their validation set is not a gold standard to compare; they should have compared it with more established data from Google. If possible, test some correlations with Google Street View data or remark this as a limitation that needs to be checked in future studies. Additionally, the English explaining the method section needs some editing as the writing is not easy! Try using proper proofreading of English before the final version of the paper. 

Reviewer 2 Report

In general, the use of street-level panoramic photos to invert road greenness is innovative, but the analysis is not deep enough.

Two other specific points to consider.

(1) "3.2. Accuracy Evaluation" shows poor improvement, and the amount(n=114) of data for accuracy evaluation is too small.

  (2) Figure 6 shows Log-log scatterplots of SGV values, while it is better to use SGV values directly.

Reviewer 3 Report

The work presented is very interesting as it proposes the possibility of using tools such as Mapillary - therefore georeferenced images collected by users with smartphones - to survey urban green areas.
The technologies used, based on machine learning, are those common to the current state of the art.
It is a very interesting reuse application, with procedures perhaps already known within the Mapillary laboratories.
The most interesting aspect, and which I believe should be underlined, is the fact that the Mapillary "component" can be replaced in the future with similar solutions and that, in any case, it is a low-cost solution where it is possible to 'hack' by introducing tools more powerful acquisition systems (e.g. 360 degree cameras).
One aspect that perhaps should be considered in the current Mapillary setup is that of data licensing: photographs are released under CC-BY-SA and, therefore, the resulting data will also need to use that license.
It's not a problem, it's still open data, just that the "share alike" constraint could create some problems in the integration of data if they are then released publicly.
Finally, since the analysis carried out does not introduce new algorithms but uses existing ones, it would be nice to find the source code (released with a special license) possibly exposed as a Jupyter Notebook on a platform like GitHub.
For instance:
in the support material there is the file imagefiltering_xgboostmodel.model on which you could show the code on how to reuse it

Reviewer 4 Report

Overall, the manuscript addresses a relevant topic, although some supplementations and improvements are required. That is why, I would like to list of major and minor remarks below.

General remarks (major):

- The author(s) use pictures (figures) without giving the source (for example figure 1 and 2). Are this photos taken by authors during the survey? Who has the rights to this photos? This issue needs to be supplemented by a certain comment.

- Literature review is not a separate section. Due to this at least a table summarizing previous studies concerning the topic should be supplemented. This would highlight the novelty of this study.

- There is no description of research area  (in section 2) or explanation why this area was chosen. For the international audience the description and location of ‘Shinjuku’ should be given as well as the spatial distribution of green coverage (on a certain map).

- Section 2.3 explains the experiment with the description of randomly chosen 200 intersections  - their locations should be given by visualising on a map – so there is no doubt that the whole area was more or less covered.   

Details (minor):

- lines 215 – 217 – a source or reference needed;

- Reference list – articles from journals concerning remote sensing or GIS analyses are in minority and there is no reference to publications previously published in IJGI, there is a doubt if this article ‘matches’ the journal topic sufficiently. I recommend a reference list supplementation.

Round 2

Reviewer 2 Report

The study can be further expanded and deepened.

Reviewer 4 Report

Thank you for your careful revisions. All of my comments from first round review were addressed. The paper has a significant improvement and the quality of the structure much higher.

However, I have 1 minor remark to your improved version of the article:

- line 157 – ‘recently’ is repeated